# Antenatal Determinants of Postnatal Renal Function in Fetal Megacystis: A Systematic Review

**DOI:** 10.3390/diagnostics14070756

**Published:** 2024-04-02

**Authors:** Ugo Maria Pierucci, Irene Paraboschi, Guglielmo Mantica, Sara Costanzo, Angela Riccio, Giorgio Giuseppe Orlando Selvaggio, Gloria Pelizzo

**Affiliations:** 1Department of Pediatric Surgery, Buzzi Children’s Hospital, 20154 Milan, Italy; ugomariapierucci@icloud.com (U.M.P.); saracostanzo@ymail.com (S.C.); angela.riccio@asst-fbf-sacco.it (A.R.); giorgio.selvaggio@asst-fbf-sacco.it (G.G.O.S.); 2Department of Biomedical and Clinical Science, University of Milano, 20157 Milan, Italy; irene.paraboschi@hotmail.com; 3Department of Surgical and Diagnostic Integrated Sciences (DISC), University of Genova, 16131 Genova, Italy; guglielmo.mantica@gmail.com

**Keywords:** fetal megacystis, lower urinary tract obstruction, posterior urethral valves, prognosis, renal function

## Abstract

**Introduction**: To evaluate the clinical usefulness of demographic data, fetal imaging findings and urinary analytes were used for predicting poor postnatal renal function in children with congenital megacystis. **Materials and methods**: A systematic review was conducted in MEDLINE’s electronic database from inception to December 2023 using various combinations of keywords such as “luto” [All Fields] OR “lower urinary tract obstruction” [All Fields] OR “urethral valves” [All Fields] OR “megacystis” [All Fields] OR “urethral atresia” [All Fields] OR “megalourethra” [All Fields] AND “prenatal ultrasound” [All Fields] OR “maternal ultrasound” [All Fields] OR “ob-stetric ultrasound” [All Fields] OR “anhydramnios” [All Fields] OR “oligohydramnios” [All Fields] OR “renal echogenicity” [All Fields] OR “biomarkers” [All Fields] OR “fetal urine” [All Fields] OR “amniotic fluid” [All Fields] OR “beta2 microglobulin” [All Fields] OR “osmolarity” [All Fields] OR “proteome” [All Fields] AND “outcomes” [All Fields] OR “prognosis” [All Fields] OR “staging” [All Fields] OR “prognostic factors” [All Fields] OR “predictors” [All Fields] OR “renal function” [All Fields] OR “kidney function” [All Fields] OR “renal failure” [All Fields]. Two reviewers independently selected the articles in which the accuracy of prenatal imaging findings and fetal urinary analytes were evaluated to predict postnatal renal function. **Results**: Out of the 727 articles analyzed, 20 met the selection criteria, including 1049 fetuses. Regarding fetal imaging findings, the predictive value of the amniotic fluid was investigated by 15 articles, the renal appearance by 11, bladder findings by 4, and ureteral dilatation by 2. The postnatal renal function showed a statistically significant relationship with the occurrence of oligo- or anhydramnion in four studies, with an abnormal echogenic/cystic renal cortical appearance in three studies. Single articles proved the statistical prognostic value of the amniotic fluid index, the renal parenchymal area, the apparent diffusion coefficient (ADC) measured on fetal diffusion-weighted MRI, and the lower urinary tract obstruction (LUTO) stage (based on bladder volume at referral and gestational age at the appearance of oligo- or anhydramnios). Regarding the predictive value of fetal urinary analytes, sodium and β2-microglobulin were the two most common urinary analytes investigated (n = 10 articles), followed by calcium (n = 6), chloride (n = 5), urinary osmolarity (n = 4), and total protein (n = 3). Phosphorus, glucose, creatinine, and urea were analyzed by two articles, and ammonium, potassium, N-Acetyl-l3-D-glucosaminidase, and microalbumin were investigated by one article. The majority of the studies (n = 8) failed to prove the prognostic value of fetal urinary analytes. However, two studies showed that a favorable urinary biochemistry profile (made up of sodium < 100 mg/dL; calcium < 8 mg/dL; osmolality < 200 mOsm/L; β2-microglobulin < 4 mg/L; total protein < 20 mg/dL) could predict good postnatal renal outcomes with statistical significance and urinary levels of β2-microglobulin were significantly higher in fetuses that developed an impaired renal function in childhood (10.9 ± 5.0 mg/L vs. 1.3 ± 0.2 mg/L, *p*-value < 0.05). **Conclusions**: Several demographic data, fetal imaging parameters, and urinary analytes have been shown to play a role in reliably triaging fetuses with megacystis for the risk of adverse postnatal renal outcomes. We believe that this systematic review can help clinicians for counseling parents on the prognoses of their infants and identifying the selected cases eligible for antenatal intervention.

## 1. Introduction

Fetal megacystis is the prenatal ultrasonographic finding of an abnormally distended bladder. It underlies a heterogeneous group of congenital anomalies of the lower urinary tract caused by a functional or mechanical bladder outlet obstruction [1].

Amongst them, posterior urethral valves (PUV) are the most common cause of this condition, existing as a spectrum of presentations in relation to the degree of the urethral obstruction and its sequelae [1].

Fetal megacystis occurs in about 3 out of 10,000 pregnancies and, if left untreated, it carries a substantial risk of perinatal mortality and long-term morbidity [2].

Perinatal death is reported in 35–55% of cases, and it is caused by the associated oligo-anydram nios that, occurring during lung development, prevents the normal canalicular phase of its embryogenesis [3,4].

End-stage kidney disease (ESKD) is described in 24–40% of survivors and can be attributed to the high pressure generated via obstruction transmitted to the upper urinary tract [2,5].

In selected cases, fetal treatment via vesicoamniotic shunting (VAS) is performed as an attempt to bypass the lower urinary tract obstruction and attenuate the secondary structural complications on the developing lungs and kidneys [3,5].

However, the success of the fetal intervention highly depends on the proper selection of suitable candidates, which is hindered by the lack of reliable prognostic factors of disease severity and progression.

Over the past few decades, many studies have investigated the use of demographic features, imaging findings, and fetal urinary analytes to prospectively predict the postnatal renal function and identify those fetuses that may benefit the most from VAS [3,4,5,6,7].

To date, however, the prognostic performances of these demographic and imaging parameters have shown contradictory results [4,7], and there is insufficient scientific evidence to recommend using fetal analytes as biomarkers of kidney damage [7].

A systematic review of the literature investigating the role of demographic characteristics, prenatal imaging findings, and fetal urinary analytes for predicting postnatal renal function in fetuses with megacystis was, therefore, performed to assist clinicians during prenatal counseling and decision-making.

## 2. Materials and Methods

This systematic review was performed according to the Preferred Reporting Items for Systematic Reviews and Metanalysis Statement (http://www.prisma-statement.org/). The PRISMA 2020 checklist. Can be downloaded at: Appendix A. 

An extensive search was conducted in the electronic database MEDLINE from inception to December 2023 using various combinations of keywords such as “luto” [All Fields] OR “lower urinary tract obstruction” [All Fields] OR “urethral valves” [All Fields] OR “megacystis” [All Fields] OR “urethral atresia” [All Fields] OR “megalourethra” [All Fields] AND “prenatal ultrasound” [All Fields] OR “maternal ultrasound” [All Fields] OR “obstetric ultrasound” [All Fields] OR “anhydramnios” [All Fields] OR “oligohydramnios” [All Fields] OR “renal echogenicity” [All Fields] OR “biomarkers” [All Fields] OR “fetal urine” [All Fields] OR “amniotic fluid” [All Fields] OR “beta2 microglobulin” [All Fields] OR “osmolarity” [All Fields] OR “proteome” [All Fields] AND “outcomes” [All Fields] OR “prognosis” [All Fields] OR “staging” [All Fields] OR “prognostic factors” [All Fields] OR “predictors” [All Fields] OR “renal function” [All Fields] OR “kidney function” [All Fields] OR “renal failure” [All Fields]. Additional records were identified by hand-searching the references reported in each selected article.

Two researchers carried out independent data extraction and quality assessment. Any disagreement was resolved by consensus or the arbitration of a third author not involved in the initial procedure. The reference lists of the selected studies and review papers were scrutinized and additional relevant articles were added. A comprehensive database was constructed using Microsoft Excel 2007 (Redmond, WA, USA).

The selection was limited to original articles, written in English, investigating the prognostic roles of demographic data, prenatal imaging findings, and fetal urinary analytes for predicting postnatal renal function in fetuses with megacystis who survived in the perinatal period.

Preclinical studies, review papers, case reports, and case series were excluded from the analysis.

Original articles investigating the prognosis of prenatally detected PUV patients were excluded from the analysis if they did not mention megacystis as the fetal uropathy identified in utero.

Similarly, retrospective studies reviewing the prenatal medical charts of postnatally diagnosed PUV patients were excluded from this study if they did not focus on cases prenatally diagnosed with megacystis.

Original articles focusing only on the efficacy of fetal surgeries were excluded from the final analysis.

Studies including fetuses who underwent termination of pregnancy (TOP) or intrauterine fetal death (IUFD) in the group of newborns with impaired renal function were excluded from the systematic review.

Similarly, original articles including neonatal deaths in the group of newborns with impaired renal function were excluded from the analysis if their kidney function was not analyzed.

Studies were also excluded from the analysis when the criteria adopted to define the postnatal renal function were not based on objective test results.

For the final analysis, for each selected article, we focused only on demographic data, prenatal imaging findings, and fetal urinary analytes of patients prenatally diagnosed with megacystis whose postnatal renal function was available at the last follow-up.

Owing to the heterogeneity and the small number of studies, a meta-analysis of the available data was deemed unfeasible.

## 3. Results

### 3.1. Overall

Out of the 727 studies that were initially identified from the main search and through the hand-searching review, 706 were excluded: 395 were excluded on a title basis, and 283 were excluded on an abstract basis. A total of 49 studies were evaluated on a full-text basis. Out of them, 20 studies were finally included in the systematic review (Figure 1).

The main features of the selected studies are reported in Table 1. Eighteen (90.0%) articles were retrospective studies, a single (5.0%) article was a prospective study, and the remaining (5.0%) article consisted of a retrospective review of 6 cases and a prospective collection of 28 cases. In total, 15 (75.0%) studies were unicentric and 5 (25.0%) studies were multicentric. A total of 13 (65.0%) studies were run by European centers (n = 6 France, n = 3 United Kingdom, n = 2 Netherlands, n = 2 Italy), 5 (25.0%) studies were run by American centers, and 2 (10.0%) studies were run by Asian centers.

Eighteen (90.0%) studies specified the study period, which ranged from 1 year to 20 years (median: 8.5 years; q_1_–q_3_: 6.3–15.0).

In total, 1049 patients were included in the 20 selected studies, with a median of 31 children (q_1_–q_3_: 23.3–80.3) per study.

Out of the 1049 patients included, in 620 (59.1%) cases, the underlying diagnosis was reported, including 462 (74.5%) cases of PUV, 62 (10.0%) cases of urethral atresia/stenosis, 25 (4.0%) cases of Prune–Belly Syndrome (PBS), 15 (2.4%) cases of vesicouretral reflux (VUR), 12 (1.9%) cases of cloacal anomalies, 6 (1.0%) cases of megalourethra, 6 (1.0%) cases of megacystis-microcolon-intestinal hypoperistalsis syndrome (MMIHS), 5 (0.8%) cases of VACTERL syndrome, 4 cases (0.6%) of unlabeled syndromes, 3 (0.5%) cases of anorectal malformations (ARM), 3 (0.5%) cases of spontaneous resolutions, 2 (0.3%) cases of trisomy 13, 2 (0.3%) cases of complex chromosomal defects, 2 (0.3%) cases of multicystic dysplastic kidney, 1 (0.2%) case of pseudo-PBS, 1 (0.2%) case of urogenital sinus, 1 (0.2%) case of anterior urethral valves, and 1 (0.2%) case of omphalocele-exstrophy-imperforate anus-spinal defect syndrome (OEIS).

Of the 1049 patients included in the 20 selected studies, 493 (47.0%) survived in the postnatal period, and the renal function measured at the last follow-up was reported together with prenatal demographic data, fetal urinary analytes, and imaging findings.

As per the selection, all studies included only fetuses prenatally diagnosed with megacystis. However, only in 4 (20.0%) articles were the criteria used to define it clearly stated. If, in 2005, Anumba et al. [8] broadly described it as an enlarged bladder that failed to empty during the maternal ultrasound (US) assessment, in 2013, Bornes et al. [9] defined it as either a bladder with a longitudinal diameter (LBD) > 7 mm in the 1st trimester of pregnancy or a bladder failing to empty over a time period of at least 45 min later in pregnancy. Similar definitions were also adopted by other authors. Duin et al. [10] used a LBD > 7 mm between the 10th and the 14th week of gestation and a bladder failing to empty during a time period of at least 40 min. Fontanella et al. [11] defined it as a LBD > 12 mm before the 18th week of gestation and an enlarged bladder failing to empty during an extended US examination lasting at least 40 min in the following gestational ages.

As shown in Table 1, the prognostic value of patients’ demographic characteristics was investigated in 10 (50.0%) articles, fetal urinary analytes in 9 (45.0%) articles, and prenatal imaging findings in 16 (80.0%) articles.

Regarding the evaluation of the postnatal renal outcomes, 11 (55.0%) articles used the serum creatinine levels, 8 (40.0%) articles used the estimated glomerular filtration rate (eGFR), and 5 (25.0%) articles used the need for renal replacement therapy (RRT).

Of the 11 studies using serum creatinine levels, 3 (27.3%) used 50 µmol/L as the cut-off value, 2 (18.2%) used 0.5 mg/dL, 1 (9.1%) used 88 µmol/L, and 1 (9.1%) used 70 mmol/L. Moreover, one (9.1%) study used a creatinine level two standard deviations lower than normal for age (without specifying the nomogram adopted), while another (9.1%) study defined a good renal outcome as a serum creatinine level of ≤ 1.0 at 1 year. Two studies did not use any specific cut-off values for serum creatinine levels but compared continuous values between two or more groups of patients: one (9.1%) study used the nadir serum creatinine level measured during the 1st year of age and one (9.1%) study used the serum creatinine level measured at regular intervals up to 1 year of age.

The eight studies using the eGFR values as postnatal renal outcomes adopted different cut-off values: Koch et al. [12] used 90 mL/min/1.73 m^2^, Bornes et al. [9] used 75 mL/min/1.73 m^2^, Duin et al. [13] used 60 mL/min/1.73 m^2^, Fontanella et al. [14] and Dreux et al. [16] used 30 mL/min/1.73 m^2^, and Moscardi et al. [14] used 15 mL/min/1.73 m^2^. According to Anumba et al. [8], a normal renal function was defined as a eGFR of 85–100 mL/min/1.73 m^2^, chronic renal failure as an eGFR of 1–84 mL/min/1.73 m^2^, and ESKD as the need for dialysis or transplantation, usually indicated by an eGFR < 10 mL/min/1.73 m^2^. Finally, Lipitz et al. [15] used an eGFR cut-off value greater than expected for the patient’s age and weight, without specifying the nomogram adopted.

### 3.2. Demographic Data

As reported in Table 2, of the 10 (50.0%) studies that evaluated the predictive role of patients’ demographic data, 8 (80.0%) investigated the gestational age (GA) at diagnosis, and 2 (20.0%) investigated the GA at delivery. While the majority of studies failed to prove their prognostic value, in 2017, Johnson et al. [16] showed that a younger GA at delivery was associated with worsening serum creatinine levels postnatally (OR—−0.1; 95% CI—−0.18, −0.03; *p* = 0.01) and the need for RRT (OR—−0.9; 95% CI—−1.5, −0.29; *p* = 0.004) postnatally.

### 3.3. Fetal Urinary Analytes

As reported in Table 3, of the nine (45.0%) studies that investigated the prognostic value of the fetal urine biochemistry, nine (100.0%) evaluated fetal urinary β2-microglobulin, nine (100.0%) sodium, five (55.6%) calcium, five (55.6%) chloride, four (44.4%) osmolarity, three (33.3%) total protein, two (22.2%) phosphorus, two (22.2%) glucose, two (22.2%) creatinine, two (22.2%) urea, one (11.1%) potassium, one (11.1%) ammonium, one (11.1%) N-Acetyl-l3-D-glucosaminidase, and one (11.1%) microalbumin. However, the cut-off values and the criteria adopted to define a normal biochemical profile varied between the studies.

While most of them failed to prove the predictive value of fetal urine biochemistry for postnatal renal function, three authors reported statistically significant results.

In the series collected by Lipitz et al. [15] in 1993, there was a statistically significant difference (*p*-value: < 0.05) in β2-microglobulin levels between the seven neonates with a prenatal diagnosis of megacystis and a confirmed postnatal diagnosis of PUV who later developed some degree of renal damage and the two neonates with PUV who maintained normal renal function at the last follow-up.

In the study published by Johnson et al. [17] in 1994, of the eight VAS survivors, the two infants with an unfavorable fetal urinary profile (sodium > 100 mg/dL, calcium > 8 mg/dL, osmolality > 200 mOsm/L, β2-microglobulin > 4 mg/L, and total protein > 20 mg/dL) were both on dialysis awaiting renal transplant, while the six fetuses with a favorable urinary biochemistry preserved intact their kidney function at the last follow-up (*p*-value: <0.05).

More recently, in 2018, Dreux et al. [13] performed a logistic regression analysis and used urine biochemistry to predict long-term renal outcomes for neonates prenatally diagnosed with megacystis and with a confirmed postnatal diagnosis of PUV. In the univariate model, fetal urine β2-microglobulin was the best single marker for the prediction of long-term renal function, showing 87% sensitivity and 72% specificity, using 5.0 mg/L as the cut-off value. Sodium and calcium were also valuable single markers, with 67% and 73% sensitivity and 85% and 65% specificity, respectively. It is worth noting that the multivariate model based on β2-microglobulin and chloride increased the sensitivity to 93% with the same specificity as β2-microglobulin alone.

### 3.4. Prenatal Imaging Findings

As reported in Table 4, of the 16 (80.0%) studies that investigated the predictive role of prenatal imaging findings, 4 (25.0%) described fetal bladder characteristics (longitudinal diameter: n = 1; wall thickness: n = 2; volume: n = 1; keyhole sign: n = 2; inability to empty: n = 1), 2 (12.5%) described ureteral findings (dilatation: n = 2), 11 (68.8%) described renal aspects, (hyperechogenicity: n = 6; cortical cysts: n = 5; hydronephrosis: n = 5; dysplasia: n = 4; parenchyma area: n = 1; anteroposterior diameter: n = 1; apparent diffusion coefficient: n = 1), and 13 (81.3%) described amniotic fluid findings (oligohydramnios: n = 6; anydramnios: n = 2; polyhydramnios: n = 1; oligo- or anydramnios: n = 1; amniotic fluid index: n = 1; amniotic fluid volume: n = 1). Moreover, Fontanella et al. [11], in 2019, developed a staging system combining both bladder and amniotic fluid findings.

With regard to bladder findings, Johnson et al. [16] showed that the inability to at least partially empty the bladder was associated with worsening renal function postnatally (OR 95% CI: −0.6 (−1.1, −0.11), *p*-value: 0.017).

When investigating the prognostic value of US imaging findings of the fetal kidneys, the hyperechogenicity of the fetal renal cortex was found to play a statistically significant role in predicting the postnatal renal function in the studies published by Duin et al. [10] and El-Ghoneimi et al. [18], renal cortical cysts in the study published by Johnson et al. [16], renal dysplasia in the study published by Anumba et al. [8] and renal parenchyma area in the study published by Moscardi et al. [14].

Moreover, in 2017, Faure et al. [19] proved that the apparent diffusion coefficient (ADC) measured during prenatal diffusion-weighted magnetic resonance imaging (MRI) was useful to predict postnatal renal function in PUV patients with a prenatal history of megacystis.

Regarding the amniotic fluid volume, despite the heterogeneity of the published literature, it was shown to have a role in predicting the postnatal renal function in the studies published by Jeong et al. [20], Faure et al. [19], Duin et al. [10], Johnson et al. [16], and Zaccara et al. [21].

Besides studies investigating single imaging findings, in 2019, Fontanella et al. [11] developed a staging system to predict the severity of the condition and its prognosis, combining the bladder volume at referral and the GA at the appearance of the associated oligo- or anhydramnios.

## 4. Discussion

VAS and other bladder drainage techniques have been described in fetuses with congenital megacystis as an attempt to ameliorate the associated pulmonary hypoplasia and increase the survival of patients with an initially poor prognosis [3,4,5].

Since the success of fetal surgery is strictly dependent on the proper selection of the most suitable candidates for fetal intervention, elucidating risk factors and measurable fetal parameters that can adversely affect postnatal renal function is fundamental to determining the most appropriate treatment options.

Hence, this systematic review aimed to collect all the literature available on prenatal determinants of postnatal renal function in fetuses with megacystis to assist clinicians during prenatal counseling and decision-making.

As per the methodology, only studies focusing on fetuses prenatally diagnosed with megacystis were eligible for inclusion. However, only 4 (20%) of the 20 articles included in the final analysis clearly stated the criteria adopted to define fetal megacystis.

As recently defined by the ERKNet CAKUT-Obstructive Uropathy Work Group [2], any fetal bladder with a LBD ≥ 7 mm in the 1st trimester of pregnancy should be considered to be affected by megacystis, and it is strongly suggestive of lower urinary tract obstruction if its LBD is ≥ 15 mm. However, no standardized cut-off values defining the degree of bladder enlargement associated with obstruction still exist in the 2nd trimester of pregnancy.

With this initial limit, a thought-out analysis of the literature available was performed to find evidence of robust predicting factors (demographic data, urinary analytes, and imaging findings) capable of assessing the severity of fetal megacystis and its long-term prognosis (Table 1).

With regard to demographic data (Table 2), the GA at the first evidence of fetal megacystis did not show any prognostic value in the eight articles investigating it. However, the GA at delivery seemed to be significantly associated with worsening postnatal renal function and the need for RRT in the study published in 2018 by Johnson et al. [16].

Regarding fetal urine assessment (Table 3), in 1993, Lipitz et al. [15] started investigating the predictive value of nine fetal urine analytes (including sodium, potassium, calcium, urea, creatinine, osmolality, β2-microglobulin, N-Acetyl-l3-D-glucosaminidase, and microalbumin) in nine fetuses with an enlarged bladder and concluded that β2-microglobulin levels were higher in those who later required peritoneal dialysis. More significant results were published 1 year later by Johnson et al. [17], who developed a specific fetal urinary signature (made of sodium, calcium, osmolarity, β2-microglobulin, and total protein) to evaluate the underlying fetal renal dysplasia and select the optimal candidates for VAS. While all six cases with a predicted good postnatal function maintained good renal outcomes, none of the two cases with an unfavorable fetal urinary signature preserved their renal function at 1 year of age. However, in 2018, Dreux et al. [13] published the first study evaluating the correlation between fetal urine biochemistry and long-term postnatal renal function (10–30 years of follow-up) in a large series of 89 cases of fetal megacystis. The authors concluded that the highest sensitivity (93%) and specificity (71%) were provided by fetal urine β2-microglobulin and chloride.

By investigating fetal imaging findings (Table 4), the assumption is usually that the larger the bladder, the more severe the underlying disease and the worse the prognosis. However, bladder imaging findings failed to show any definitive value for the prediction of the postnatal renal function, except for fetal inability to empty the bladder at least partially. This was probably due to the lack of defined normal ranges for fetal bladder size and urine production.

While several authors showed the predictive value of the US aspects of the fetal kidneys (including the hyperechogenicity of the renal cortex, the presence of renal cortical cysts, and renal dysplasia in the renal parenchyma area), in 2017, Faure et al. [19] first described the roles of ADC values, as measured by using diffusion-weighted imaging (DWI)-MRI, as non-invasive tools for postnatal renal function prediction. The authors concluded that MRI was accurate for the evaluation of renal parenchyma, and the addition of diffusion sequence DW-MRI helped estimate future renal function. Fetal DWI-MRI with ADC determination can, therefore, be considered an additional method for renal assessment when biologic and US findings are inconclusive, especially in the case of oligohydramnios.

Since, from approximately 18 weeks of gestation, the amniotic fluid is predominately driven by the excretion of fetal urine, and its volume has been investigated by several authors to evaluate the severity of the underlying urinary obstruction and predict the prognosis of the affected fetuses, albeit with contradictory results.

In this regard, the most remarkable study included in the analysis was performed by eight academic hospitals in the Netherlands, and it was published in 2019 [11]. The authors classified the condition into three stages of severity (mild, moderate, severe) based on bladder volume at referral and GA at the appearance of the associated oligo- or anyhdramnios. They proved that postnatal renal function in children prenatally diagnosed with megacystis progressively worsened with the severity of the stage of the underlying disease.

Besides the amniotic fluid volume and the GA at the appearance of the associated oligo- or anyhdramnios, peptidomics analyses have been more recently performed on the amniotic fluid to identify specific peptide-based signatures capable of assessing the disease severity and predicting the postnatal renal function [31]. Similar metabolic analyses have also been performed on fetal urine [32]. Very promising results have been achieved, and an international prospective and case-control study including 400 patients is currently ongoing for extensive clinical validation [33].

The strengths of this study included the fact that it conducted a thorough systematic review of the literature, encompassing a wide range of articles to evaluate the predictive value of demographic data, fetal imaging findings, and urinary analytes in children with congenital megacystis. The systematic search strategy employed in this study ensured a comprehensive and unbiased selection of relevant articles, enhancing the reliability of the findings. By analyzing various demographic, imaging, and urinary parameters, this study provided a comprehensive assessment of potential predictors for postnatal renal function, offering valuable insights for clinical practice. This study’s findings have direct implications for clinical decision-making, particularly in counseling parents on the prognosis of infants with congenital megacystis and identifying cases suitable for antenatal intervention. Despite the heterogeneous nature of the included studies, the identification of specific urinary biomarkers associated with postnatal renal outcomes represents a significant contribution to the field, potentially guiding future research and clinical management.

However, the variability in study designs, populations, and methodologies among the included articles may have introduced heterogeneity and limited the generalizability of the findings. Despite investigating a range of urinary analytes, this study found limited evidence supporting their prognostic value for postnatal renal function, highlighting the need for further research in this area. The reliance on the published literature may introduce publication bias, as studies reporting statistically significant findings are more likely to be published, potentially skewing the overall results. This study encountered challenges in synthesizing data due to the lack of standardized outcome measures across studies, which may have influenced the interpretation of results.

Despite the constraints inherent in this study, it highlights the importance of demographic data, fetal imaging parameters, and urinary analytes in triaging fetuses with megacystis for the risk of adverse postnatal renal outcomes. Addressing these limitations and considering them in the interpretation of this study’s findings will be crucial for ensuring the accuracy and applicability of the research in clinical practice and future research endeavors.

We envisage that, in the years to come, demographical data, urinary analytes, and imaging findings will be integrated with a specific signature derived from the maternal and fetal fluid samples using complementary analytical tools, such as bioinformatics and machine learning techniques, to develop an algorithm capable of predicting the postnatal renal function of fetuses with megacystis and improve the decision-making capabilities of current antenatal prediction models.

## 5. Conclusions

With the limits of the heterogeneity of the definition adopted and the different study designs and eligibility criteria used in the studies included in the final analysis, several demographic data, fetal urinary analytes, and imaging parameters were shown to play roles in reliably triaging fetuses with megacystis for the risk of adverse postnatal renal outcomes. We believe that this systematic review can help clinicians when counseling parents on the prognosis of their infants and identifying the selected cases eligible for antenatal intervention.

## Figures and Tables

**Figure 1 diagnostics-14-00756-f001:**
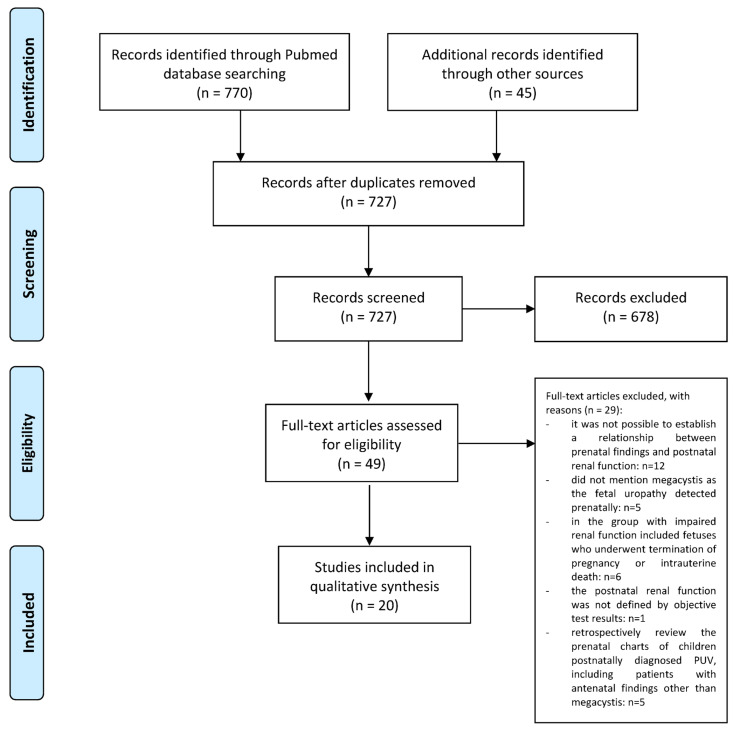
Flow Diagram.

**Table 1 diagnostics-14-00756-t001:** Original articles included in the systematic review investigating demographic data, prenatal imaging findings, and fetal urinary analytes for the prediction of postnatal renal function in fetuses diagnosed with megacystis.

Author, Year	Study Design	Study Period	Target Population	Number of Patients Included in the Study	Underlying Diagnosis	Number of Patients Included in the Systematic Review	Prenatal Prognostic Factors Investigated			Timing of Renal Function Evaluation
							Demographic Data	Prenatal Imaging Findings	Fetal Urinary Analytes	
Anumba et al. [8], 2005	Retrospective multicentric study	1984–1997	Fetuses with an enlarged bladder failing to empty during the prenatal ultrasound assessment	113	PUVs n = 72; urethral atresia n = 33; PBS n = 4; unknown = 4	15	no	yes	no	18 months of age
Bornes et al. [9], 2013	Retrospective unicentric study	1989–2009	Fetuses prenatally diagnosed with megacystis	84	PUVs n = 25; urethral atresia n = 10; congenital megalourethra n = 3; VUR n = 9; trisomy 18 n = 7; trisomy 13 n = 2; complex chromosomal defects n = 2; anorectal malformations n = 3; cloacae n = 2; urogenital sinus n = 1; VACTERL associations n = 5; MMIHS n = 4; OEIS n = 1; PBS n = 1; unlabeled syndromes n = 4; spontaneous resolution n = 5	32	yes	no	no	1 year of minimum follow-up
Duin et al. [10], 2019	Retrospective multicentric study	2000–2015	Fetuses with a prenatal diagnosis of megacystis and a confirmed postnatal diagnosis of LUTO	95	nd	95	yes	yes	no	1st year after diagnosis
Fontanella et al. [11], 2019	Retrospective multicentric study	2007–2014	Fetuses with early/late megacystsis at a high risk of isolated LUTO	261	nd	66	no	yes	no	1st year after birth
Koch et al. [12], 2021	Retrospective unicentric study	1994–2013	Fetuses with isolated LUTO (defined as megacystsis with a dilated posterior urethra and bilateral hydronephrosis) undergoing 2 sequential urine analyses	26	PUVs n = 17; megalourethra n = 2; PBS n = 2; urethral atresia/stenosis n = 2; multicystic kidney dysplasia n = 2; VUR n = 1	9	no	no	yes	5 years of age
Dreux et al. [13], 2018	Retrospective multicentric study	1986–2005	Fetuses with a prenatal diagnosis of megacystis in the 2nd–3rd trimesters of pregnancy and a confirmed postnatal diagnosis of PUVs	89	PUVs n = 89	89	no	no	yes	Minimum of 10 years of age
Moscardi et al. [14], 2018	Retrospective unicentric study	2009–2015	Fetuses with megacystsis who survived to birth following VAS	15	PUVs n = 8; PBS n = 4; urethral atresia n = 2; MMIHS n = 1	15	yes	yes	no	1 year of age
Lipitz et al. [15], 1993	Retrospective unicentric study	nd	Fetuses with LUTO (as defined by an enlarged bladder)	25	PUVs n = 15	8	no	yes	yes	Last follow-up
Johnson et al. [16], 2018	Retrospective multicentric study	2007–2012	Fetuses prenatally diagnosed with LUTO (as defined by an enlarged bladder with dilated proximal urethra and the presence of hydronephrosis with pyelicocaliectasis) with normal midgestational amniotic fluid volume	32	PUVs n = 18; PBS n = 8; VUR n = 4; urethral stricture n = 1; pseudo-PBS n = 1	25	yes	yes	no	Newborn discharge, 1-year follow-up, and 2-year follow-up
Johnson et al. [17], 1994	Retrospective unicentric review (n = 6)/prospective analysis (n = 28)	nd	Fetuses diagnosed with LUTO (described as megacystis, bilateral hydronephrosis and a decreased amniotic fluid volume)	34	TOP n = 11; spontaneous resolution n = 3; IUFD n = 4; neonatal death n = 1; cloacal anomalies n = 2; PUVs n = 3; anterior urethral valves and megalourethra n = 1; MMIHS n = 1; urethral atresia n = 3; PBS n = 5	8	no	yes	yes	1 year of age
El-ghoneimi et al. [18], 1999	Retrospective unicentric study	1989–1996	Fetuses prenatally diagnosed with PUVs (as defined by megacystis and severe bilateral hydronephrosis)	30	PUVs n = 30	9	yes	yes	yes	Last follow-up (median: 3 years; q_1–_q_3_: 2.0–6.0 years)
Faure et al. [19], 2017	Retrospective unicentric study	2003–2014	Fetuses with suspected PUVs on the basis of the ultrasonographic finding of severe urinary tract anomalies (oligohydramnios, hyperechoic kidneys, megacystis and/or bilateral hydronephrosis) undergoing fetal MRI	11	PUVs n = 9; TOP n = 2	9	yes	yes	no	1st year after birth
Jeong et al. [20], 2018	Retrospective unicentric study	1998–2013	Fetuses with LUTO (as defined by an enlarged urinary bladder and a dilated proximal urethra) undergoing VAS	32	PUVs n = 15; cloacal anomaly n = 7; urethral stenosis n = 3; TOP n = 5; IUFD n = 2	22	no	yes	no	28 days after birth and 2 years of age
Zaccara et al. [21], 2005	Retrospective unicentric study	1999–2002	Fetuses with a distended and thickened bladder and enlarged kidneys	28	Urethral atresia n = 6; PUVs n = 19; unknown n = 3	18	no	yes	no	1st year after birth
Hutton et al. [22], 1997	Retrospective unicentric study	1982–1992	Fetuses prenatally diagnosed with PUVs (as defined by a dilated urinary bladder and upper tract dilatation)	32	PUVs n = 32	13	no	yes	no	Last follow-up (median: 5.7 years; range: 4.4– 10 years)
Won et al. [23], 2006	Retrospective unicentric study	1998–2005	Fetuses with bilateral hydronephrosis associated with evidence of bladder outlet obstruction and oligohydramnios	8	PUVs n = 5; cloacal anomaly n = 1; urethral stenosis n = 1; IUFD = 1	6	yes	yes	yes	Last follow-up (median: 1.9 years; q_1–_q_3_: 0.5–3.2 years)
Craparo et al. [24], 2007	Retrospective unicentric study	1999–2006	Fetuses with megacystis and oligohydramnios	12	PUVs n = 12	10	yes	no	yes	Last follow-up (median: 1.5 years; q_1–_q_3_: 1.0–1.8 years)
Sarhan et al. [25], 2008	Retrospective unicentric study	1987–2004	Fetuses with prenatally detected PUVs (as defined by megacystis and bilateral severe hydronephrosis)	79	PUVs n = 79	15	yes	yes	yes	Last follow-up (median: 9.6 years; q_1–_q_3_: 6.4–10.8 years)
Ruano et al. [26], 2016	Retrospective unicentric study	2013–2014	Fetuses with primary LUTO (as defined by a dilated urinary bladder associated with bilateral hydroureter/hydronephrosis)	25	nd	15	yes	yes	yes	6 months of age
Nassr e al [27], 2019	Retrospective unicentric study	2013–2017	Fetuses with LUTO (as defined by an enlarged urinary bladder with hydronephrosis and/or hydroureter) and a normal amniotic fluid volume in the 2nd trimester of pregnancy	18	PUVs n = 14; VUR n = 1; congenital bulbar urethral stricture n = 1; PBS n = 1; anterior urethral valve n = 1	14	no	yes	no	2 years of age

Abbreviations. LUTO: lower urinary tract obstruction; PUVs: posterior urethral valves; VAS: vesicoamniotic shunting; PBS: Prune–Belly syndrome; VUR: vesicoureteral reflux; MMIHS: megacystis-microcolon-intestinal hypoperistalsis syndrome; TOP: termination of pregnancy; IUFD: intrauterine fetal death; OEIS: omphalocele-exstrophy-imperforate anus-spinal defect syndrome; nd: not defined.

**Table 2 diagnostics-14-00756-t002:** Original articles included in the systematic review investigating the role of demographic data for the prediction of postnatal renal function in fetuses with megacystis.

Author, Year	Threshold	Criteria Adopted for Defining Renal Function	Timing of Renal Function Evaluation	Results
**GA at diagnosis**				
Bornes et al. [9], 2013	2nd trimester vs. 3rd trimester	Good renal function: eGFR > 75 mL/min/1.73 m^2^	1 year of minimum follow-up	Good renal function: 7/8 (87.5%) patients diagnosed in the 2nd trimester vs. 23/24 (95.8%) patients diagnosed in the 3rd trimester (*p*-value: ns)
Duin et al. [10], 2019	na	Compromised renal function: eGFR < 60 mL/min/1.73 m^2^ based on the nadir serum creatinine level	1st year after diagnosis	OR (95% CI): 0.971 (0.917–1.027); *p*-value: 0.305
Dreux et al. [13], 2018	25 weeks	Poor renal function: eGFR < 30 mL/min/1.73 m^2^	Minimum 10 years of age	AUC (95% CI): 0.44 (0.33–0.55)
Moscardi et al. [14], 2018	na	ESRD: eGFR < 15 mL/min/1.73 m^2^	1 year of age	Mean (±sd): 18.17 (±2.93) weeks in patients with ESRD vs. 18.92 (±3.37) weeks in non-ESRD patients (*p*-value: ns)
El-Ghoneimi et al. [18], 1999	na	Normal renal function: serum creatinine level < 50 µmol/L (0.56 mg/dL)	Last follow-up (median: 3 years; q_1_–q_3_: 2.0–6.0 years)	Median: 33 weeks (q_1_–q_3_: 24–33 weeks) in patients with normal renal function vs. 33 weeks (q_1_–q_3_: 29–34 weeks) in patients with impaired renal function (*p*-value: ns)
Faure et al. [19], 2017	3rd trimester of gestation	Nadir serum creatinine level	1st year after birth	Median nadir serum creatinine level: 24 μmol/L (q_1–_q_3_: 23–48 μmol/L) in patients diagnosed during the 3rd trimester of pregnancy vs. 49 μmol/L (q_1––_q_3_: 30–65 μmol/L) in patients diagnosed before the 3rd trimester of pregnancy (*p*-value: ns)
Sarhan et al. [25], 2008	na	Normal renal function: serum creatinine level < 50 μmol/L (0.56 mg/dL)	Last follow-up: median 9.6 years (q_1_–q_3_: 6.4–10.8 years)	Median: 28.5 weeks (q_1_–q_3_: 23–33 weeks) in patients with normal renal function vs. 29 weeks (q_1_–q_3_: 25–33 weeks) in patients with impaired renal function (*p*-value: ns)
Ruano et al. [26], 2016	na	Normal renal function: serum creatinine level < 0.5 mg/dL	6 months of age	OR (95% CI): 0.92 (0.73–1.13); Pr (OR < 1) = 22.5%
Ruano et al. [26], 2016	1st trimester	Normal renal function: serum creatinine level < 0.5 mg/dL	6 months of age	OR (95% CI): 0.91 (0.06–10.5); Pr (OR > 1) = 47.3%
**GA at delivery**				
Johnson et al. [16], 2018	na	Worsening serum creatinine level ≥ 0.51 mg/dL	Newborn discharge, 1-year follow-up, 2-year follow-up	Univariate analysis: OR (95% CI): −0.1 (−0.2; −0.1); *p*-value: <0.0001
				Multivariate analysis: OR (95% CI): −0.1 (−0.18, −0.03); *p*-value: 0.01)
Johnson et al. [16], 2018	na	Need for RRT	Newborn discharge, 1-year follow-up, 2-year follow-up	Univariate analysis: OR (95% CI): −1.1 (−1.7; −0.4); *p*-value: <0.001
				Multivariate analysis: OR (95% CI): −0.9 (−1.5; −0.29); *p*-value: 0.004)
Craparo et al. [24], 2007	na	Renal insufficiency NOS	Last follow-up (median: 1.5 years; q_1_–q_3_: 1.0–1.8 years)	Median: 38 weeks (q_1_-q_3_: 37.5–38.5 weeks) in patients with normal renal function vs. 36.5 weeks (q_1_–q_3_: 36–38 weeks) in patients with impaired renal function (*p*-value: ns)

Abbreviations. GA: gestational age; eGFR: estimated glomerular filtration rate; ESRD: end-stage renal disease; RRT: renal replacement therapy; OR: odds ratio; CI: confidence interval; AUC: area under the Receiver Operating Characteristic (ROC) curve; q_1_–q_3_: interquartile range; sd: standard deviation; na: not applicable; vs.: versus; NOS: not otherwise specified.

**Table 3 diagnostics-14-00756-t003:** Original articles included in the systematic review investigating the role of urine biochemistry for the prediction of the postnatal renal function in fetuses diagnosed with megacystis.

Author, Year	Fetal Urinary Analytes/Profiles Investigated	GA at Sampling (Weeks)	Criteria Adopted for Renal Function Evaluation	Timing of Renal Function Evaluation	Results
Koch et al. [12], 2021	Normal biochemical profile: sodium < 100 mmol/L, chloride < 90 mmol/L, calcium < 8 mg/dL, β2-microglobulin < 6 mg/L on the 1st ultrasound-guided bladder puncture; improving biochemical profile: decrease in at least one of the above parameters on the 2nd ultrasound-guided bladder puncture performed 24–48 h after the 1st one	nd	Normal renal function: eGFR > 90 mL/min/1.73 m^2^	5 years of age	Normal renal function: 3/3 (60.0%) with normal biochemical profile vs. 5/6 (83.3%) with improving biochemical profile (*p*-value: ns)
Dreux et al. [13], 2018	β2-microglobulin > 5.0 mg/L; phosphorus > 0.2 mmol/L; protein > 0.05 g/L; sodium > 60 mmol/L; calcium > 1.2 mmol/L; chloride > 60 mmol/L; glucose > 0.2 mmol/L; (β2-microglobulin × chloride)/10 > 18; (β2-microglobulin × sodium × calcium)/100 > 3	Median: 32 weeks (range: 21–36 weeks)	Poor renal function: eGFR < 30 mL/min/1.73 m^2^	Minimum 10 years of age	Univariate model AUC (95% CI): β2-microglobulin: 0.85 (0.67–0.90); phosphorus: 0.72 (0.58–0.82); protein: 0.73 (0.51–0.79); sodium: 0.72 (0.63–0.89); calcium: 0.69 (0.48–0.74); chloride: 0.72 (0.55–0.82); glucose: 0.73 (0.63–0.85)
					Multivariate model AUC (95% CI): (β2-microglobulin × chloride)/10: 0.89 (0.81–0.96); (β2-microglobulin × sodium × calcium)/100: 0.79 (0.75–0.95)
Lipitz et al. [15], 1993	Sodium (mmol/L); potassium (mmol/L); calcium (mmoI/L); urea (µmol/L); creatinine (µmol/L); osmolality (mOsm/L); β2-microglobulin (mg/L); N-Acetyl-l3-D-glucosaminidase (U/L); microalbumin (mg/L) **	nd	Impaired renal function: eGFR greater than expected for their age and weight or serum creatinine level > 70 mmol/L after the 1st week of life or need for peritoneal dialysis	nd	Mean sodium level (±sd): 65.5 (±12) mmol/L with normal renal function vs. 81.3 (±23) mmol/L with impaired renal function (*p*-value: ns)
Johnson et al. [17], 1994	Favorable prognostic criteria: sodium < 100 mg/dL; calcium < 8 mg/dL; osmolality < 200 mOsm/L; β2-microglobulin < 4 mg/L; total protein < 20 mg/dL)	nd	Normal renal function: serum creatinine level ≤ 1.0	1 year of age	Normal renal function: 6/6 (100.0%) with favorable prognostic criteria vs. 0/2 (0.0%) with unfavorable prognostic criteria (*p*-value: <0.05)
El-Ghoneimi et al. [18], 1999	Sodium; chloride; calcium: phosphorus; glucose; ammonium; creatinine; urea; total protein; β2-microglobulin *	nd	Normal renal function: serum creatinine level < 50 µmol/L (0.56 mg/dL)	Last follow-up (median: 3 years; q_1_–q_3_: 2.0–6.0 years)	Mean microalbumin level (±sd): 25 (± 27) mg/L with normal renal function vs. 48 (± 33) mg/L with impaired renal function (*p*-value: ns)
Won et al. [23], 2006	Favorable prognostic criteria: sodium < 100 mmol/L; chloride < 90 mmol/L; osmolarity < 200 mOsm; β2-microglobulin (<4 mg/L	Median: 19.1 weeks (IQR: 17–21.6 weeks)	Normal renal function: serum creatinine level < 88 mmol/L	Last follow-up (median: 1.9 years; q_1_–q_3_: 0.5–3.2 years)	Mean N-Acetyl-l3-D-glucosaminidase level (±sd): 53 (± 1 U/L) with normal renal function vs. 190 (±86 U/L) with impaired renal function (*p*-value: ns)
Craparo et al. [24], 2007	Sodium > 100 mEq/L; β2-microglobulin > 13 mg/L	Median: 23 weeks (IQR: 20–28 weeks)	Renal insufficiency NOS	Last follow-up (median: 1.5 years; q_1_–q_3_: 1.0–1.8 years)	Mean osmolality (±sd): 146 (±23) mOsm/L with normal renal function vs. 181 (±23) mOsm/L with impaired renal function (*p*-value: ns)
Sarhan et al. [25], 2008	Sodium (mmol/L); β2-microglobulin (mg/L)	Median: 33 weeks (IQR: 31–33 weeks)	Normal renal function: serum creatinine level < 50 µmol/L (0.56 mg/dL)	Last follow-up: median 9.6 years (q_1_–q_3_: 6.4–10.8 years)	Mean urea level (±sd): 4.5 (±0.0) µmol/L with normal renal function vs. 5.6 (±2.0) µmol/L with impaired renal function (*p*-value: ns)
Ruano et al. [26], 2016	Favorable urinary biochemistry: sodium < 100 mEq/L; chloride < 90 mEq/L; osmolarity < 200 mOsm/L; β2-microglobulin < 6 mg/L	Range: 18–30 weeks	Normal renal function: serum creatinine level < 0.5 mg/dL	6 months of age	Mean creatinine level (±sd): 181 (±99) µmol/L with normal renal function vs. 180 (±74) µmol/L impaired renal function (*p*-value: ns)
					Mean calcium level (±sd): 0.8 (±0.1) mmol/L with normal renal function vs. 1.2 (±0.7) mmol/L with impaired renal function (*p*-value: ns)

Abbreviations. GA: gestational age; eGFR: estimated glomerular filtration rate; AUC: area under the Receiver Operating Characteristic (ROC) curve; CI: confidence interval; IQR: interquartile range; sd: standard deviation; ns: not statistically significant; nd: not disclosed; vs.: versus; NOS: not otherwise specified. * Based on a cut-off level correlated with gestational age, depending on the reference [28]. ** Values were compared with previously published reference ranges [29].

**Table 4 diagnostics-14-00756-t004:** Original articles included in the systematic review investigating the role of prenatal imaging findings in the prediction of postnatal renal function in fetuses with megacystis.

Parameter	Author	Threshold	Criteria Adopted to Define the Renal Function	Timing of Renal Function Evaluation	Results
**Bladder findings**					
Bladder longitudinal diameter	Duin et al. [10], 2019	na (mm)	Compromised renal function: eGFR < 60 mL/min/1.73 m^2^ based on the nadir serum creatinine level	1st year after diagnosis	OR (95% CI): 0.998 (0.971–1.025); *p*-value: 0.865
Bladder wall thickness	Duin et al. [10], 2019	na (mm)	Compromised renal function: eGFR < 60 mL/min/1.73 m^2^ based on the nadir serum creatinine level	1st year after diagnosis	OR (95% CI): 1.037 (0.937–1.147); *p*-value: 0.477
Bladder wall thickened	Duin et al. [10], 2019	y/n	Compromised renal function: eGFR < 60 mL/min/1.73 m^2^ based on the nadir serum creatinine level	1st year after diagnosis	OR (95% CI): 0.431 (0.111–1.665); *p*-value: 0.221
Bladder volume	Fontanella et al. [11], 2019	na (cm^3^)	Severely impaired renal function: eGFR < 30 mL/min/1.73 m^2^ based on the nadir serum creatinine level	1st year after birth	Univariate analysis: OR (95% CI): 1.0 (0.97–1.04); *p*-value: 0.89
					Multivariate analysis: OR (95% CI): 1.0 (0.96–1.04), *p*-value: 0.99
Keyhole sign	Duin et al. [10], 2019	y/n	Compromised renal function: eGFR < 60 mL/min/1.73 m^2^ based on the nadir serum creatinine level	1st year after diagnosis	OR (95% CI): 2.645 (0.800–8.333); *p*-value: 0.111
	Moscardi et al. [14], 2018	na (cm)	ESRD: eGFR < 15 mL/min/1.73 m^2^	1 year of age	Mean 3.66 (±2.82) cm in patients with ESRD vs. 3.77 (±3.18) cm in non-ESRD patients (*p*-value: ns)
Inability to empty the bladder (at least partially)	Johnson et al. [16], 2018	y/n	Worsening serum creatinine level ≥ 0.51 mg/dL	Newborn discharge, 1-year follow-up, and 2-year follow-up	OR (95% CI): −0.6 (−1.1, −0.11); *p*-value: 0.017
	Johnson et al. [16], 2018	y/n	Need for RRT	Newborn discharge, 1-year follow-up, and 2-year follow-up	OR (95% CI): −1.4 (−2.8, 0.1); *p*-value: 0.059
**Ureteral findings**					
Ureteral dilatation	Fontanella et al. [11], 2019	na (mm)	Severely impaired renal function: eGFR < 30 mL/min/1.73 m^2^ based on the nadir serum creatinine level	1st year after birth	Univariate analysis: OR (95% CI): 1.17 (0.98–1.39); *p*-value: 0.81
	Ruano et al. [26], 2016	y/n	Normal renal function: serum creatinine level < 0.5 mg/dL	6 months of age	Multivariate analysis: OR (95% CI): 1.12 (0.92–1.37); *p*-value: 0.27
					OR (95% CI): 0.17 (0.01–3.93); Pr (OR > 1): 84.9%
**Renal findings**					
Hyperechogenicity of the renal cortex	Duin et al. [10], 2019	y/n	Compromised renal function: eGFR < 60 mL/min/1.73 m^2^ based on the nadir serum creatinine level	1st year after diagnosis	Univariable logistic regression: OR (95% CI): 2.647 (1.041–6.734); *p*-value: 0.041
	El-Ghoneimi et al. [18], 1999	y/n	Normal renal function: serum creatinine level < 50 µmol/L (0.56 mg/dL)	Last follow-up (median: 3 years; q_1_–q_3_: 2.0–6.0 years)	Normal renal function: 1/3 (33.3%) in patients with hyperechoic kidneys vs. 3/3 (100.0%) in patients without hyperechoic kidneys (*p*-value: ns) Normal renal function: 3/3 (100.0%) with in patients with hyperechoic kidneys vs. 2/6 (33.3%) in patients without hyperechoic kidneys (*p*-value: < 0.05)
	Faure et al. [19], 2017	y/n	Nadir serum creatinine level	1st year after birth	Multivariable logistic regression: OR (95% CI): 2.433 (0.934–6.338); *p*-value: 0.069
	Won et al. [23], 2006	y/n	Normal renal function: serum creatinine level < 88 mmol/L	Last follow-up (median: 1.9 years; q_1_–q_3_: 0.5–3.2 years)	Normal renal function: 2/6 (33.3%) in patients with hyperechoic kidneys vs. 2/9 (22.2%) in patients without hyperechoic kidneys (*p*-value: ns)
	Sarhan et al. [25], 2008	y/n	Normal renal function: serum creatinine level < 50 μmol/L (0.56 mg/dL)	Last follow-up: median 9.6 years (q_1_–q_3_: 6.4–10.8 years)	OR (95% CI): 4.35 (0.53–44.12); Pr (OR > 1): 91.5%
	Ruano et al. [26], 2016	y/n	Normal renal function: serum creatinine level < 0.5 mg/dL	6 months of age	Median nadir serum creatinine level: 49 μmol/L (q_1_–q_3_: 30–169 μmol/L in patients with hyperechoic kidneys vs. 25 μmol/L (q_1_–q_3_: 24–48 μmol/L) in patients without hyperechoic kidneys (*p*-value: ns)
Renal cortical cysts	Johnson et al. [16], 2018	y/n	Worsening serum creatinine level ≥ 0.51 mg/dL	Newborn discharge, 1-year follow-up, and 2-year follow-up	OR (95% CI): 0.55 (0.1–1.1), *p*-value: 0.045
		y/n	Need for RRT	Newborn discharge, 1-year follow-up, and 2-year follow-up	OR (95% CI): 1.9 (0.1–3.8), *p*-value: 0.041
	El-Ghoneimi et al. [18], 1999	y/n	Normal renal function: serum creatinine level < 50 µmol/L (0.56 mg/dL)	Last follow-up (median: 3 years; q_1_–q_3_: 2.0–6.0 years)	Normal renal function: 0/1 (0.0%) patients with renal cortical cysts vs. 5/8 (62.5%) patients without renal cortical cysts (*p*-value: ns)
	Won et al. [23], 2006	y/n	Normal renal function: serum creatinine level < 88 mmol/L	Last follow-up (median: 1.9 years; q_1_–q_3_: 0.5–3.2 years)	Normal renal function: 0/1 (0.0%) patients with renal cortical cysts vs. 4/5 (80.0%) patients without renal cortical cysts (*p*-value: ns)
	Sarhan et al. [25], 2008	y/n	Normal renal function: serum creatinine level < 50 μmol/L (0.56 mg/dL)	Last follow-up: median 9.6 years (q_1_–q_3_: 6.4–10.8 years)	0/4 (0.0%) patients with normal renal function vs. 5/11 (45.5%) patients with impaired renal function (*p*-value: ns)
	Ruano et al. [26], 2016	y/n	Normal renal function: serum creatinine level < 0.5 mg/dL	6 months of age	OR (95% CI): 7.50 (0.22–405.19); Pr (OR > 1): 86.6%
Renal dysplasia	Anumba et al. [8], 2005	Echogenic e/o cystic kidneys (y/n)	Impaired renal function: eGFR < 85 mL/min/1.73 m^2^	18 months of age	Impaired renal function: 6/6 (100.0%) patients with echogenic e/o cystic kidneys vs. 3/9 (33.3%) with normal kidneys (*p*-value: <0.05)
	Anumba et al. [8], 2005	Echogenic e/o cystic kidneys (y/n)	ESRD: eGFR < 10 mL/min/1.73 m^2^	18 months of age	ESRD: 1/6 (16.7%) patients with echogenic *e*/*o* cystic kidneys vs. 1/9 (11.1%) with normal kidneys (*p*-value: ns)
	Fontanella et al. [11], 2019	Abnormal renal cortical appearance NOS (y/n)	Severely impaired renal function: eGFR < 30 mL/min/1.73 m^2^ based on the nadir serum creatinine level	1st year after birth	Univariate analysis: OR (95% CI): 3.3 (0.54–20.23); *p*-value: 0.19
	Moscardi et al. [14]	Unilateral cystic kidney and/or echogenic renal parenchyma in the 2nd trimester of pregnancy (y/n)	ESRD: eGFR < 15 mL/min/1.73 m^2^	1 year of age	Multivariate analysis: OR (95% CI): 2.96 (0.21–42.66); *p*-value: 0.43
	Moscardi et al. [14]	Cystic kidney and/or echogenic renal parenchyma in the 3rd trimester of pregnancy (y/n)	ESRD: eGFR < 15 mL/min/1.73 m^2^	1 year of age	5/8 (50.0%) patients with ESRD vs. 1/7 (14.3%) patients with non-ESRD (*p*-value: ns)
	Ruano et al. [26], 2016	Enlarged hyperechogenic kidney with no corticomedullary differentiation and small cysts in the cortex (y/n)	Normal renal function: serum creatinine level < 0.5 mg/dL	6 months of age	OR (95% CI): 0.98 (0.00–135.33), Pr (OR > 1): 50.3%
Abnormal kidney ADC value	Faure et al. [19], 2017	ADC value > 1.8 mm^2^s^−1^ (y/n)	Nadir serum creatinine level	1st year after birth	Median nadir serum creatinine level: 57 μmol/L (q_1_-q_3_: 39.5–117 μmol/L) in patients with abnormal ADC value vs. 24.5 μmol/L (q_1_-q_3_: 23.5–36.5μmol/L) in patients with normal ADC value (*p*-value: < 0.05)
Renal parenchyma area	Moscardi et al. [14], 2018	na (cm^2^)	ESRD: eGFR < 15 mL/min/1.73 m^2^	1 year of age	AUC (95% CI): 0.793 (0.639–0.949)
Renal anteroposterior diameter	Duin et al. [10], 2019	na (mm)	Compromised renal function: eGFR < 60 mL/min/1.73 m^2^ based on the nadir serum creatinine level	1st year after diagnosis	OR (95% CI): 0.990 (0.936–1.047); *p*-value: 0.719
Hydronephrosis	Duin et al. [10], 2019	Renal pelvis DAP (mm)	Compromised renal function: eGFR < 60 mL/min/1.73 m^2^ based on the nadir serum creatinine level	1st year after diagnosis	OR (95% CI): 0.698 (0.197–2.482); *p*-value: 0.579
	El-Ghoneimi et al. [18], 1999	Hydronephrosis NOS (y/n)	Normal renal function: serum creatinine level < 50 µmol/L (0.56 mg/dL)	Last follow-up (median: 3 years; q_1_–q_3_: 2.0–6.0 years)	Normal renal function: 5/8 (62.5%) patients with hydronephrosis vs. 0/1 (0.0%) patients without hydronephrosis (*p*-value: ns)
	Hutton et al. [22], 1997	Megacystis alone or associated with mild upper tract dilatation (DAP: 5–9 mm) vs. megacystis associated with moderate/severe upper tract dilatation (DAP: >10 mm)	Impaired renal function: serum creatinine level 2 standard deviations higher than normal for age or need of RRT	Last follow-up (median: 5.7 years; range: 4.4–10 years)	Impaired renal function: 2/8 (25.0%) patients with megacystis alone or associated with mild upper tract dilatation (DAP: 5–9 mm) vs. 4/5 (80.0%) patients with megacystis associated with moderate/severe upper tract dilatation (DAP: >10 mm) (*p*-value: ns)
	Sarhan et al. [25], 2008	At least monolateral hydronephrosis NOS (y/n)	Normal renal function: serum creatinine level < 50 μmol/L (0.56 mg/dL)	Last follow-up: median 9.6 years (q_1_–q_3_: 6.4–10.8 years)	Normal renal function: 4/13 (30.8%) patients with hydronephrosis vs. 0/2 (0.0%) patients without hydronephrosis (*p*-value: ns)
	Ruano et al. [26], 2016	Grade ≥ 2 according to Grignon’s classification * (y/n)	Normal renal function: serum creatinine level < 0.5 mg/dL	6 months of age	OR (95% CI): 2.05 (0.09–72.69); Pr (OR > 1) = 67.3%
**Amniotic fluid findings**					
Oligohydramnios	Duin et al. [10], 2019	SDP < 3 cm (y/n)	Compromised renal function: eGFR < 60 mL/min/1.73 m^2^ based on the nadir serum creatinine level	1st year after diagnosis	Univariable logistic regression: OR (95% CI): 2.074 (0.705–6.099); *p*-value: 0.185
	Moscardi et al. [14], 2018	AFI < 5 cm or < 5th percentile for the corresponding GA at delivery (y/n)	ESRD: eGFR < 15 mL/min/1.73 m^2^	1 year of age	Oligohydramnios: 7/8 (87.5%) patients with ESRD vs. 4/7 (57.1%) non-ESRD patients (*p*-value: ns)
	Lipitz et al. [15], 1993	Amniotic fluid volume NOS	Impaired renal function: eGFR greater than expected for age and weight or serum creatinine level > 70 mmol/L after the 1st week of life or need for peritoneal dialysis	nd	Impaired renal function: 1/2 (50%%) patients with normal amniotic fluid volume, 3/4 (75.9%) patients with mild–moderate oligohydramnios, and 3/3 (100.0%) patients with severe oligohydramnios (*p*-value: ns)
	Faure et al. [19], 2017	Oligohydramnios NOS (y/n)	Nadir serum creatinine level	1st year after birth	Median nadir serum creatinine level: 117 μmol/L (q_1_–q_3_: 65–169 μmol/L) in patients with oligohydramnios NOS vs. 27.5 μmol/L (q_1_–q_3_: 24–48 μmol/L) in patients without oligohydramnios (*p*-value: < 0.05)
	Jeong et al. [20], 2018	SDP < 2 cm at diagnosis (mild oligohydramnios: SFD: 1–2 cm; severe oligohydramnios: SDP < 1 cm)	Normal renal function: serum creatinine level of < 50 μmol/L	28 days after birth	Normal renal function: no severe oligohydramnios (*p*-value < 0.05)
	Nassr et al. [27], 2019	AFI < 5th percentile adjusted for GA, in the 3rd trimester (y/n)	Need for RRT	2 years of age	Multivariable logistic regression: OR (95% CI): 2.451 (0.863-6.962); *p*-value: 0.092
					Need for RRT: 3/8 (37.5%) patients with oligohydramnios vs. 1/6 (16.7%) patients with normal AF volume (*p*-value: ns)
Anhydramnios	Duin et al. [10], 2019	y/n	Compromised renal function: eGFR < 60 mL/min/1.73 m^2^ based on the nadir serum creatinine level	1st year after diagnosis	Univariable logistic regression: OR (95% CI): 12.116 (1.284–117.074); *p*-value: 0.029
					OR (95% CI): 3.01 (0.38–33.28); Pr (OR > 1): 85.1%
	Ruano et al. [26], 2016	y/n	Normal renal function: serum creatinine level < 0.5 mg/dL	6 months of age	OR (95% CI): 3.01 (0.38–33.28); Pr (OR > 1): 85.1%
Polyhydramnion	Duin et al. [10], 2019	SDP > 8 cm (y/n)	Compromised renal function: eGFR < 60 mL/min/1.73 m^2^ based on the nadir serum creatinine level	1st year after diagnosis	Univariable logistic regression: OR (95% CI): 0.990 (0.936–1.047); *p*-value: 0.795
					Multivariable logistic regression: OR (95% CI): 2.256 (0.215–23.672); *p*-value: 0.497
Oligohydramnios or anhydramnios	Anumba et al. [8], 2005	SDP ≤ 2 cm (y/n)	Impaired renal function: eGFR < 85 mL/min/1.73 m^2^	18 months of age	Impaired renal function: 5/7 (71.4%) patients with oligohydramnios or anhydramnios vs. 4/8 (50.0%) without oligohydramnios or anhydramnios (*p*-value: ns)
	Anumba et al. [8], 2005	SDP ≤ 2 cm (y/n)	ESRD: eGFR < 10 mL/min/1.73 m^2^	18 months of age	ESRD: 2/7 (28.6 patients with oligohydramnios or anhydramnios vs. 0/8 (0.0%) patients without oligohydramnios or anhydramnios (*p*-value: ns)
	Johnson et al. [16], 2018	Oligohydramnios or anhydramnios NOS (y/s)	Worsening serum creatinine level ≥ 0.51 mg/dL	2 years follow-up	Univariate analysis: OR (95% CI): 0.8 (0.4–1.2); *p*-value: < 0.0001
	Johnson et al. [16], 2018	Oligohydramnios or anhydramnios NOS (y/s)	Need for RRT	2 years follow-up	Multivariate analysis: OR (95% CI): 0.5 (0.07–0.96); *p*-value: 0.023
					Univariate analysis: OR (95% CI): 2.5 (0.9–4.2); *p*-value: 0.003
					Multivariate analysis: OR (95% CI): OR (95% CI) 1.7 (−0.3, 3.7); *p*-value: 0.09)
	El-Ghoneimi et al. [18], 1999	Oligohydramnios or anhydramnios NOS (y/n)	Normal renal function: serum creatinine level < 50 µmol/L (0.56 mg/dL)	Last follow-up (median: 3 years; q_1_–q_3_: 2.0–6.0 years)	Normal renal function: 2/3 (66.7%) with oligo- or anhydramnios vs. 3/6 (50.0%) patients without oligohydramnios or anhydramnios (*p*-value: ns)
	Sarhan et al. [25], 2008	Oligohydramnios or anhydramnios NOS (y/s)	Normal renal function: serum creatinine level < 50 μmol/L (0.56 mg/dL)	Last follow-up: median 9.6 years (q_1_–q_3_: 6.4–10.8 years)	Normal renal function: 3/9 (33.3%) patients with oligohydramnios or anhydramnios vs. 1/6 (16.7%) with patients without oligohydramnios or anhydramnios (*p*-value: ns)
AFI	Moscardi et al. [14], 2018	na	ESRD: eGFR < 15 mL/min/1.73 m^2^	1 year of age	3.17 (±2.22) at delivery in patients with ESRD vs. 11.93 (±11.33) at delivery in patients with non-ESRD: (*p*-value: ns)
	Zaccara et al. [21], 2005	AFI < 25th percentile vs. AFI between the 50th and 75th percentile	Serum creatinine level	1st year of age	Mean serum creatinine level: 1.3 (±0.2) mg/dL in patients with AFI < 25th percentile vs. 0.6 (±0.1) mg/dl in patients with AFI between the 50th and 75th percentile (*p*-value < 0.05)
Amniotic fluid volume	Dreux et al. [14], 2018	na (mL)	Poor renal function: eGFR < 30 mL/min/1.73 m^2^	Minimum 10 years of age	AUC (95% CI): 0.32 (0.23–41)
**Staging systems**					
LUTO stage	Fontanella et al. [11], 2019	Severe LUTO stage: bladder volume ≥ 5.4 cm^3^ and/or oligo- or anhydramnios before 20 weeks; moderate LUTO stage: bladder volume < 5.4 cm^3^ and/or normal AF volume at 20 weeks; mild LUTO stage: normal AF volume at 26 weeks	Severely impaired renal function: eGFR < 30 mL/min/1.73 m^2^ based on the nadir serum creatinine level	1st year after birth	Impaired renal function: 4/9 (44.4%) patients with severe LUTO stage; 5/16 (31.3%) patients with moderate LUTO stage; 4/36 (11.1%) patients with mild LUTO stage (*p*-value: < 0.05)

Abbreviations. GA: gestational age; eGFR: estimated glomerular filtration rate; SDP: single deepest pocket; AFI: amniotic fluid index; LUTO: lower urinary tract obstruction; ADC: apparent diffusion coefficient; DAP: anteroposterior diameter; ESRD: end stage renal disease; RRT: renal replacement therapy; OR: odds ratio; CI: confidence interval; AUC: aera under the Receiver Operating Characteristic (ROC) curve; not statistically significant; na: not applicable; NOS: not otherwise specified; ns: y/n: yes/no. * Reference [30].

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
