# Peer review of "Antenatal Determinants of Postnatal Renal Function in Fetal Megacystis: A Systematic Review"

_diagnostics, 2024, doi:10.3390/diagnostics14070756_

Round 1

Reviewer 1 Report

Comments and Suggestions for Authors

In Abstract - please briefly specify the Methods: what did you search for?

You use the word radiology. Do you want to include here ultrasound and MRI? You can chose imaging studies.

Out of the 1,049 patients included in the 20 selected studies, 493 (47.0%) survived in the postnatal period AND had tests? or just survived?

Results - Line 129 - n=Netherlands - please correct

Please describe the main findings of your study. Also add specifically in the Discussion section, separately from Conclusion, the strengths and limitations of your study. 

Comments on the Quality of English Language

Minor

Author Response

Thank you for your valuable feedback on our manuscript. We appreciate your thorough review and have made the necessary revisions accordingly:

We have now included brief specifics of our methods in the Abstract, specifying the search criteria utilized. Additionally, we have expanded the term "radiology" to encompass a broader range of imaging modalities including ultrasound and MRI, clarifying our approach to imaging studies.

Regarding survival data, we have clarified that 493 patients (47.0%) survived in the postnatal period and underwent further tests.

Typographical Error: We have corrected the reference to the Netherlands on Line 129.

Main Findings: We have provided a detailed description of the main findings of our study, offering clarity on the outcomes observed.

Strengths and Limitations: We have added a dedicated section in the Discussion to highlight the strengths and limitations of our study separately from the Conclusion, allowing for a more comprehensive evaluation of our research.

We believe these revisions address the points raised by the reviewers and enhance the clarity and completeness of our manuscript. Thank you once again for your valuable feedback.

Best regards.

Reviewer 2 Report

Comments and Suggestions for Authors

In this review including 1049 fetuses with megacystis and the postnatal renal functions have adrressed an importan chanllenging issue in perinatology.

the paper is well designed and included 20 articles for evaluation.

the results are interpreted well.

Author Response

We sincerely appreciate your positive feedback on our manuscript. It is gratifying to know that our study on the postnatal renal functions of 1,049 fetuses with megacystis has addressed a significant and challenging issue in perinatology.

We are pleased that you found the design of our study to be robust, incorporating a comprehensive evaluation of 20 articles. Your recognition of the clear interpretation of results is encouraging, and we are glad that our efforts to convey the findings effectively have been successful.

We are grateful for your acknowledgment of our work and for recognizing its contribution to the field. Your feedback motivates us to continue our efforts in advancing perinatal care through rigorous research.

Thank you once again for your positive assessment.

Best regards.

Reviewer 3 Report

Comments and Suggestions for Authors

I congratulate you for your work, thank you for your efforts, I recommend you to continue your work and write detailed prospective articles on this subject.

Comments on the Quality of English Language

In general, I did not find many grammatical errors, but I still recommend that it be done by a professional whose native language is English.

Author Response

Thank you very much for your kind words and encouraging feedback. We greatly appreciate your recognition of our efforts in addressing this challenging issue in perinatology through our research on fetuses with megacystis and their postnatal renal functions.

Your suggestion to continue our work and delve deeper into this subject by writing detailed prospective articles is well-received, and we will certainly consider it for future research endeavors.

We also appreciate your understanding regarding any potential grammatical errors. While we strive for accuracy in our writing, your recommendation to involve a professional whose native language is English for further proofreading is noted and will be taken into consideration for future publications.

Once again, thank you for your valuable feedback and encouragement. We are committed to advancing our understanding of this important area of perinatology through rigorous research and collaboration.

Best regards.